# Accuracy of prenatal screening for congenital heart disease in population: A retrospective study in Southern France

Cornélie Suard[1]*, Audrey Flori[2], Florent Paoli[3], Anderson Loundou[4], Virginie Fouilloux[5], Sabine Sigaudy[6,7], Fabrice Michel[8], Julie Antomarchi[9], Pamela Moceri[10], Véronique Paquis-Flucklinger[11], Claude D'Ercole[1,12,13,14‡], Florence Bretelle[1,12,13,14‡]

1 Department of Gynecology and Obstetrics, Nord Hospital, Assistance Publique des Hôpitaux de Marseille (AP-HM), Aix Marseille Université, Marseille, France, 2 Department of Gynecology, Nice Teaching Hospital, Nice, France, 3 Pediatric Cardiology Service, Assistance Publique des Hopitaux de Marseille, Hôpital de la Timone Enfants, Marseille, France, 4 Department of Public Health, Medical Evaluation, Assistance Publique des Hôpitaux de Marseille, Aix- Marseille Université, Marseille France, 5 Department of Cardiac Surgery, Children's Hospital La Timone, Assistance Publique des Hôpitaux de Marseille, Marseille, France, 6 Department of Medical Genetics, Children's Hospital La Timone, Assistance Publique des Hopitaux de Marseille, Marseille, France, 7 CPDPN Timone-Conception, Marseille, France, 8 Department of Pediatric Intensive Care Unit, Assistance-Publique des Hôpitaux de Marseille, Hospital La Timone, Marseille, France, 9 Department of Obstetrics and Gynecology, Nice Teaching Hospital, Nice, France, 10 Department of cardiology, Nice Teaching Hospital, Nice, France, 11 Department of Medical Genetics, National Center for Mitochondrial diseases, Nice Teaching Hospital, Nice, France, 12 Department of Obstetrics and Gynecology, Conception hospital, Assistance Publique des Hôpitaux de Marseille (AP-HM), Aix Marseille Université, Marseille, France, 13 CPDPDN Timone Conception. Réseau Méditerranée (PACA Corse Monaco), Marseille, France, 14 IHU, IRD, Marseille, France

☯ These authors contributed equally to this work.
‡ These authors also contributed equally to this work.
* cornélie.suard@gmail.com

**Data Availability Statement:** All relevant data are within the manuscript and its Supporting Information files.

## Abstract

Congenital heart diseases (CHDs) are the most common congenital malformations. The objective of our study was to evaluate the prenatal screening accuracy of congenital heart disease (CHD) in Southern France and to evaluate the impact of a prenatal diagnosis on pregnancies outcomes and neonatal outcomes. We performed a bicentric, retrospective observational study in the southern region over 4 years was conducted between 1 January 2014 and 31 December 2017. All foetuses and children under one year of age with CHD monitored in the UTHs (University Teaching Hospitals) in Marseille and Nice were included. CHD cases were divided into 3 groups: group 1, those with no possible options for anatomical repair; group 2, those with anatomical repair possibilities but that may require neonatal cardiologic management; and group 3, those with anatomical repair possibilities that do not require an emergency neonatal procedure. Among the 249070 deliveries during the study period, 677 CHD cases were included in the study. The overall prenatal screening rate was 71.5%. The screening rates were 97.8%, 63.6%, and 65.9% for groups 1, 2 and 3, respectively. Among group 2 CHD cases, 80% of the transpositions of the great arteries, 56% of the aortic coarctations, and 20% of the total anomalous pulmonary venous returns were detected during the prenatal period. A genetic anomaly was found in 16% of CHD cases.

**Funding:** The author(s) received no specific funding for this work.

**Competing interests:** The authors have declared that no competing interests exist.

The overall mortality rate was 11.3% with a higher death rate in cases of prenatal screening (17.2% versus 2.1%; p < 0.001). However, when focusing only on children who died of CHD, prenatal screening did not create an impact (56.6% versus 100%, p = 0,140). Our data showed that the prenatal screening rate of CHD appears satisfactory in Southern France. Nevertheless, it could be improved for some CHD. This study did not find any benefit in terms of mortality from prenatal screening for CHD.

## Introduction

Congenital heart diseases (CHDs) are the most common congenital malformations, representing one-third of all cases [1]. In Europe, these malformations represent approximately thirty-six thousand live births per year, or a prevalence of approximately seven in one thousand live births [1,2]. CHD is a major cause of mortality due to congenital malformation in the first year of life [3].

In European countries, the screening rate varies depending on the modalities of the screening programmes. The screening rate ranges from 17.9% in the absence of organised screening to 55.6% when 2 or 3 ultrasounds are carried out systematically [4]. In France, three ultrasounds are recommended during pregnancy.

However, in 2016, the National Perinatal Survey showed that an average of 5.5 ultrasounds were performed [5]. This screening rate varies depending on the region, from 47.3% in the Paris area [6] to 71% in Haute-Normandie [7].

The benefits of prenatal CHD screening regarding morbidity-mortality appear inconsistent in terms of survival according to some studies [4,6–9]. However, among neonates with CHD, a prenatal diagnosis seems to be associated with lower rates of preoperative risk factors for cardiac surgery [10]. Furthermore, a prenatal diagnosis could improve the prognosis of children with regard to morbidity, particularly concerning the neurocognitive development level [9,11,12]. Some CHDs are associated with genetic abnormalities, and prenatal screening for them allows for a genetic investigation [13,14].

Some studies have demonstrated the value of training on the efficiency of prenatal heart disease screening [7,15].

Prenatal screening for CHD has never been studied in the southern region of France.

The main objective of this study was to assess prenatal screening rates for CHD in Southern France and their impact on pregnancies outcomes and neonatal outcomes.

## Materials and methods

This study specifically received approval from the Ethics Committee at the University of Aix-Marseille on May 29, 2018 (file reference: 2018-24-05-007). The CIL (Correspondant Informatique et Libertés) was made aware of the study complying with French law (reference: DSN_2018-07-27_7419). Data were collected from anonymized medical records.

A 4-year retrospective observational study was carried out in Southern France between 1 January 2014 and 31 December 2017. This regional study included the University Teaching Hospitals (UTHs) in Marseille (UTH Nord and UTH Timone) and Nice (UTH Archet and UTH Lenval) as well as their three Multidisciplinary Prenatal Diagnosis Centres (MPDCs) located at the UTH Nord, the UTH Timone in Marseille, and the UTH Archet in Nice.

The data were collected retrospectively from computerised patient medical files by two researchers and anonymized. Research was performed on Viewpoint ultrasound software (GE healthcare) to help identify all foetuses with CHD. An inventory of all the children hospitalised

in one of the departments was performed using the ICD-10 (International Classification of Diseases) Code for CHD.

## Inclusion criteria

This study included foetuses and children under the age of one year whose diagnosis of CHD was made in their prenatal or postnatal period during the study period. All foetuses with CHD appraised in one of the 3 MPDCs involved during the study period and all children under the age of one year with CHD discovered during the first year of life who needed hospitalisation in the paediatric cardiology, paediatric cardiac surgery or paediatric cardiac intensive care units at the UTH in Marseille or in the paediatric department at the UTH in Nice during the study period were included.

The CHD cases studied were classified into 3 groups according to the classification system used in the study by Durand et al. [7] (Fig 1):

- Group 1: a heart defect with no possibility for anatomical repair.

- Group 2: a heart defect with a possibility for anatomical repair but that may require neonatal cardiologic management.

- Group 3: a heart defect with the possibility for anatomical repair that does not require emergency neonatal procedures.

Double aortic arches, ventricular septal defects, heart tumours and anomalies of the origin of the pulmonary artery were also considered. The exclusion criteria were heart rhythm disorders, isolated pericardial effusion, patent ductus arteriosus, atrial septal defects and anomalies at the origins of the coronary arteries. Anatomic variation such as right aortic arch isolated were not recorded.

## Data collected

For each foetus included, the data collected were the gestational age at diagnosis, the existence of polymalformative syndrome (PMS) or an associated genetic anomaly, the occurrence of foetal death, and the realisation of a termination of pregnancy (TOP) with an analysis of the gestational age and aetiology of it. In cases of a postnatal diagnosis, the parameters studied were the age at the time of diagnosis and the discovery of an associated PMS or genetic anomaly. For all children born with CHD, the following data were collected: the requirement for hospitalisation in the neonatal intensive care unit and the duration in number of days, the existence of PMS or an associated genetic anomaly, and the occurrence of death.

## Definitions

PMS is defined as the presence of another organ malformation or foetal growth restriction associated with CHD. Complex cardiopathy is defined as CHD that cannot be repaired and is associated with multiple anomalies in the cardiac structure that do not make it possible to classify it in any one category according to its anatomy. Anomalies in the number or structure of the chromosomes grouped together under the term "genetic anomalies". Amniocentesis with karyotype or CGH array was proposed for all patient in case of prenatal diagnosis.

## Statistical analysis

Statistical analysis was performed using PASW Statistics version 17.02 (IBM SPSS, Inc., Chicago, IL, USA). Continuous variables are expressed as means ± SDs or as medians with ranges

**GROUP 1: A heart defect with no possible anatomical repair options**
- o   Hypoplasia of the left ventricle
- o   Single ventricle
- o   Tricuspid atresia
- o   Complex cardiomyopathy †
- o   Endocardial fibroelastosis

**GROUP 2: A heart defect with the possibility for anatomical repair but may require neonatal cardiologic management**
- o   Transposition of the great arteries
- o   Coarctation of the aorta/Aortic stenosis/Interruption of the aortic arch
- o   Shone's syndrome
- o   Pulmonary atresia with VSD, of which 3 were associated with double discordance
- o   Pulmonary atresia with intact VS and critical pulmonary stenosis
- o   Agenesis of the pulmonary valves
- o   TAPVR #
- o   Double aortic arch

**GROUP 3: A heart defect with the possibility for anatomical repair that does not require an emergency neonatal procedure**
- o   Atrioventricular canal
- o   Tetralogy of Fallot
- o   Common arterial trunk
- o   Aortopulmonary window
- o   Tricuspid valve dysplasia and Ebstein's anomaly
- o   Double discordance
- o   Tight pulmonary stenosis
- o   Ventricular septal defects (VSD)  ‡
- o   Aneurysm of the right ventricle
- o   Heart tumour
- o   Anomaly in the origin of the pulmonary artery

† CHD that cannot be repaired and associated with several anomalies of the cardiac structure that do not make it possible to classify them in any category according to their anatomy . # Total anomalous pulmonary venous return. ‡ Divided into peri-membranous, admission, muscular and conoventricular VSD, VSD: ventricular septal defect

**Fig 1. Classification of congenital heart disease types.** † CHD that cannot be repaired and associated with several anomalies of the cardiac structure that do not make it possible to classify them in any category according to their anatomy. # Total anomalous pulmonary venous return. ‡ Divided into peri-membranous, admission, muscular and conoventricular VSD, VSD: ventricular septal defect.

(min, max), and categorical variables are reported as counts and percentages. Comparisons of the mean values between two groups were performed using Student's t-test or the Mann-Whitney U test. Comparisons of percentages were performed using a Chi-square test or Fisher's exact test, as appropriate. All tests were two-sided, and statistical significance was defined as $p < 0.05$.

## Results

### Population

Among 249070 deliveries during the study period, 772 CHD cases were identified. Seventy-one cases were excluded because the data on pregnancy outcomes were incomplete, and 24

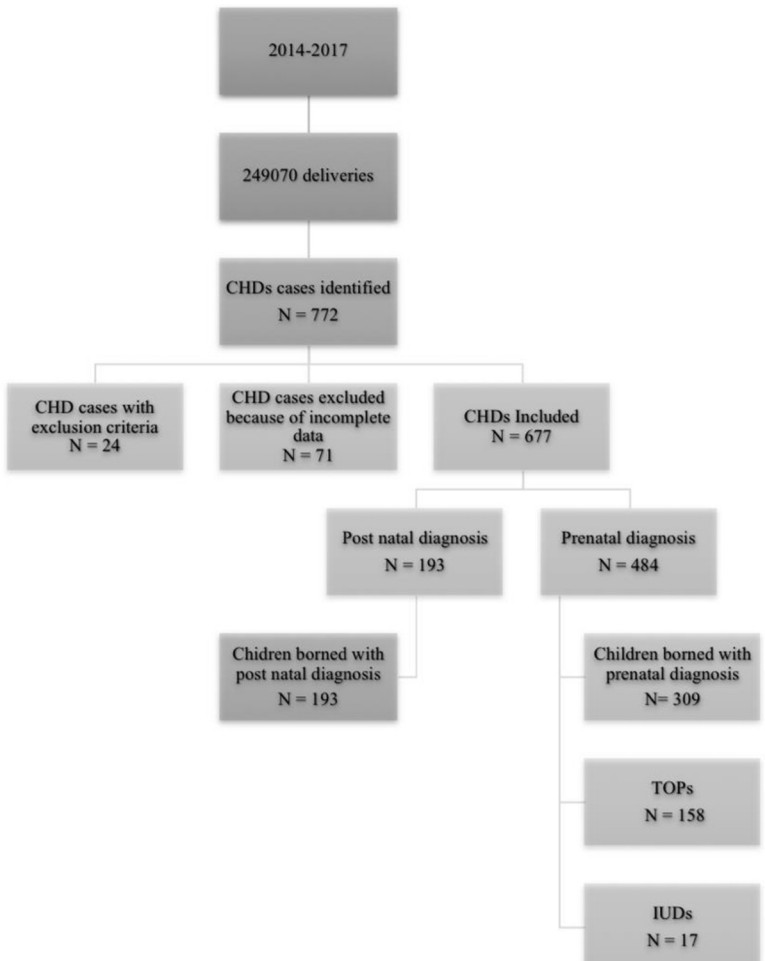

**Fig 2. Flow chart. CHD**: Congenital heart disease. **TOP**: Terminations of pregnancy; **IUD**: Intrauterine demises.

more were excluded because they did not meet the inclusion criteria (Fig 2). In total, 677 CHD cases were included, out of which 134 (19.8%) were classified into group 1, 206 (30.4%) into group 2 and 337 (49.8%) into group 3. The most commonly identified CHDs were ventricular septal defects (21.3%), tetralogy of Fallot (10.5%), coarctations of the aorta (10.5%) and atrio-ventricular septal defects (9.5%) (Table 1). The screening rates per year were 69.2%, 68.2%, 82.9% and 71.1%, respectively, from 2014 to 2017.

## Prenatal screening

A total of 484 CHD cases were identified in the prenatal period, with an overall prenatal detection rate of 71.5%. The detection rate after excluding ventricular septal defects was 77.1%. The detection rates for each CHD studied are described in Table 1. The proportion of CHD cases detected in utero varied by group, with a detection rate of 97.8% in group 1, 63.6% in group 2 and 65.9% in group 3 (Table 1). The average gestational age upon diagnosis of CHD in prenatal cases was 23.7 weeks of gestation (+/- 5.4). A total of 76.6% of CHD cases diagnosed in the prenatal period were detected in the second trimester of pregnancy, 18.3% were detected during the third trimester. In our study, only 5.1% of CHDs were diagnosed in the first trimester. Among the 25 CHDs diagnosed in the first trimester, thirteen were from group 1, one from

**Table 1. Proportion of prenatal screening of congenital heart disease among the 3 groups.**

| CHD | Prenatal diagnosis<br>n (%) | Postnatal diagnosis<br>n (%) | Total<br>n (%) |
|---|---|---|---|
| GROUP 1 | | | |
| Left ventricle hypoplasia | 52 (98.1) | 1 (1.9) | 53 (7.8) |
| Single ventricle | 51 (98.1) | 1 (1.9) | 52 (7.7) |
| Tricuspid atresia | 11 (100) | 0 | 11 (1.6) |
| Complex cardiomyopathy | 16 (94.1) | 1 (5.9) | 17 (2.5) |
| Endocardial fibroelastosis | 1 (100) | 0 | 1 (0.1) |
| **Subtotal group 1** | 131 (97.8) | 3 (2.2) | 134 (19.8) |
| GROUP 2 | | | |
| Transposition of the great arteries | 42 (80.8) | 10 (19.2) | 52 (7.7) |
| Coarctation of the aorta | 40 (56.3) | 31 (43.7) | 71 (10.5) |
| Aortic stenosis | 6 (35.3) | 11 (64.7) | 17 (2.5) |
| Shone's syndrome | 0 | 1 (100) | 1 (0.1) |
| PA-VSD † | 10 (83.3) | 2 (16.7) | 12 (1.8) |
| PA-IVS ‡ and critical pulmonary stenosis | 14 (77.8) | 4 (22.2) | 18 (2.7) |
| Interruption of the aortic arch | 2 (40) | 3 (60) | 5 (0.7) |
| Agenesis of the pulmonary valves | 5 (100) | 0 | 5 (0.7) |
| TAPVR § | 3 (21.4) | 11 (78.6) | 14 (2.1) |
| Double aortic arch | 9 (81.8) | 2 (18.2) | 11 (1.6) |
| **Subtotal group 2** | 131 (63.6) | 75 (36.4) | 206 (30.4) |
| GROUP 3 | | | |
| Atrioventricular septal defect | 59 (92.2) | 5 (7.8) | 64 (9.5) |
| Tetralogy of Fallot | 53 (74.6) | 18 (25.4) | 71 (10.5) |
| Common arterial trunk | 7 (77.8) | 2 (22.2) | 9 (1.3) |
| Aortopulmonary window | 0 | 2 (100) | 2 (0.3) |
| Tricuspid valve dysplasia and Ebstein's anomaly | 11 (91.7) | 1 (8.3) | 12 (1.8) |
| Double discordance | 4 (100) | 0 | 4 (0.8) |
| Tight pulmonary stenosis | 6 (28.6) | 15 (71.4) | 21 (3.1) |
| Ventricular septal defect | 73 (50.7) | 71 (49.3) | 144 (21.3) |
| Heart tumour | 7 (100) | 0 | 7 (1) |
| Anomaly of the origin of the PA ¶ | 1 (50) | 1 (50) | 2 (0.3) |
| Aneurysm of the right ventricle | 1 (100) | 0 | 1 (0.1) |
| **Subtotal group 3** | 222 (65.9) | 115 (34.1) | 337 (49.8) |
| **Total** | 484 (71.5) | 193 (28.5) | 677 (100) |

† Pulmonary atresia with a ventricular septal defect.

‡ Pulmonary atresia with an intact ventricular septum.

§ Total anomalous pulmonary venous return.

¶ PA: pulmonary arteries

group 2, eleven from group 3. Regarding the outcome of these pregnancies: eighteen TOPs were performed at an average gestational age 16.2 weeks, three IUDs were observed, and four children were born and had surgery.

## Postnatal screening

In total, 28.5% CHD cases were not identified in the prenatal period (N = 193). A total of 57.5% of these cases were discovered in the first week of life. The median age at diagnosis, all

cases taken together, was 5 days (2–30). The median age at diagnosis was 1 day (1–90), 4 days (1–18) and 7.5 days (3–30) for cases in groups 1, 2 and 3, respectively.

## Polymalformative syndromes and genetic anomalies

Amongst the 677 CHD cases identified, PMS was found in 21.8% of cases (N = 148) and a genetic anomaly was detected in 16.1% of cases (N = 109). PMS and genetic anomalies were more common in group 3 (65.5% and 70.6%, respectively). The rates of PMS and genetic anomalies were significantly higher in cases of prenatal diagnosis than in cases of postnatal diagnosis (24.8% versus 14.5%; $p < 0.05$ and 18% versus 11%; $p < 0.05$, respectively). These differences were particularly significant for group 3 (33.8% versus 19.1%; $p < 0.05$ for PMS and 27% versus 14.8%; $p < 0.05$ for genetic abnormalities). The main genetic anomalies identified were trisomy 21 (35.7%), trisomy 18 (19.5%) and microdeletion 22Q11 (15.6%). The other anomalies found were trisomy 13, triploidy, deletion chromosome 4,5,6,7,8,10, 17,18, Turner syndrome, duplication chromosome X, duplication chromosome 8, partial trisomy chromosome 11 and 22.

## Outcomes

**Evolution of the pregnancies in cases of prenatal screening (Table 2).**   Among the 484 CHD cases identified in the prenatal period, 309 (63.8%) pregnancies resulted in the birth of a living child. Seventeen (3.6%) cases of intrauterine demise were observed, and 158 (32.6%) cases resulted in a TOP. Of the 158 TOP cases, 74 (46.8%) were carried out for CHD cases in group 1, 17 (10.8%) for CHD cases in group 2 and 67 (42.4%) for CHD cases in group 3. The TOP procedures carried out for CHD cases in groups 1 and 2 were performed because of the severity of the CHD in 64 (86.5%) and 12 (70.6%) cases, respectively. For group 3, a TOP was carried out due to a genetic cause in 36 (53.7%) cases. An autopsy was performed in 35.4% of cases of TOPs and in 11.7% of cases of IUDs. Among the children borned alive with a CHD diagnosed prenatally, only thirteen diagnoses were reversed. These thirteen cases were prenatal suspicions of coarctation of the aorta that did not occur at birth. All other diagnosis were postnatally confirmed.

**Outcomes for children with CHD (Table 3).**   A total of 502 children were born with CHD. In 309 (61.5%) cases, the diagnosis was prenatal, and in 193 (38.5%) cases, the diagnosis was postnatal. Among the children borned alive with a CHD diagnosed prenatally, all the diagnosis were postnatally confirmed. Hospitalisation in neonatal intensive care was more often

**Table 2. Outcome of pregnancies with congenital heart disease according to a prenatal diagnosis.**

|  | Group 1 N (%) | Group 2 N (%) | Group 3 N (%) | Total N (%) |
|---|---|---|---|---|
| Live births | 46 (35.1) | 112 (85.5) | 151 (68) | 309 (63.8) |
| TOP | 74 (56.5) | 17 (13) | 67 (30.2) | 158 (32.6) |
| GA at TOP | 22.7 | 28.8 | 24.8 | 24.2 |
| TOP Cause: |  |  |  |  |
| - Severity of CHD | 64 (86.5) | 12 (70.6) | 15 (22.4) | 91 (57.6) |
| - PMS | 5 (6.8) | 2 (11.8) | 16 (23.9) | 23 (14.6) |
| - Genetic anomaly § | 5 (6.8) | 3 (17.6) | 36 (53.7) | 44 (27.8) |
| Intrauterine demise | 11 (8.4) | 2 (1.5) | 4 (1.8) | 17 (3.6) |
| Total prenatal diagnosis | 131(100) | 131(100) | 222 (100) | 484 (100) |

**TOP** = Termination of pregnancy; **GA** = Gestational age; **PMS** = Polymalformative syndrome

**§** This term includes anomalies of the karyotype, anomalies of the CGH array and Mendelian genetic syndromes

**Table 3. Outcomes of children with congenital heart disease.**

| | CHD with a prenatal diagnosis | CHD without a prenatal diagnosis | p |
|---|---|---|---|
| **Group 1 N = 49** | 46 | 3 | |
| Hospitalisation in the neonatal intensive care unit N(%) | 43 (93,5) | 2 (66.7) | 0,230 |
| Average duration of hospitalisation in intensive care (days) (± ET) | 23 (± 31,3) | 24 (± 31) | NS |
| Deaths N (%) | 25 (54.3) | 1 (33.3) | 0,594 |
| Cause of death: cardiopathy | 22 (88) | 1 (100) | 0.999 |
| Death before surgery | 11 (50) | 1 (100) | 0,999 |
| Cause of death: other cause # | 3 (12) | 0 (0) | 0.999 |
| **Group 2 N = 187** | 112 | 75 | |
| Hospitalisation in the neonatal intensive care unit N(%) | 103 (92) | 61 (81,3) | 0,01 |
| Average duration of hospitalisation in intensive care (days) (± ET) | 15.4 (±14,9) | 15.1 (±11) | NS |
| Deaths N (%) | 12 (10,7) | 3 (4) | 0,109 |
| Cause of death: cardiopathy | 5 (41,7) | 3 (100) | 0,200 |
| Death before surgery | 4 (80) | 2 (66.7) | 0.999 |
| Cause of death: other cause # | 7 (58,3) | 0 (0) | 0,200 |
| **Group 3 N = 266** | 151 | 115 | |
| Hospitalisation in the neonatal intensive care unit N(%) | 53 (35,1) | 19 (16.5) | < 0.001 |
| Average duration of hospitalisation in intensive care (days) (± ET) | 18.6 (±27,2) | 18.2 (±22,1) | NS |
| Deaths N (%) | 16 (10,6) | 0 (0) | < 0,001 |
| Cause of death: cardiopathy | 3 (18.8) | 0 (0) | NA |
| Death before surgery | 3 (100) | 0 | NA |
| Cause of death: other cause # | 13 (81.2) | 0 (0) | NA |
| **Total population N = 502** | 309 | 193 | |
| Hospitalisation in the neonatal intensive care unit N(%) | 200 (64,7) | 82 (42.5) | < 0.001 |
| Average duration of hospitalisation in intensive care (days) (± ET) | 17.8 (±22,8) | 15.9 (± 14,3) | 0,051 |
| Deaths N (%) | 53 (17,2) | 4 (2.1) | < 0.001 |
| Cause of death: cardiopathy | 30 (56,6) | 4 (100) | 0,140 |
| Death before surgery | 17 (56.7) | 3 (75) | 0.627 |
| Cause of death: other causes # | 23 (43,4) | 0 (0) | 0,140 |

# Other causes of death: genetics anomalies, polymalformative syndrome, prematurity, infection

found in prenatal diagnosis cases than in postnatal diagnosis cases (64.7% versus 42.5%; p < 0,001). No significant difference was found in terms of hospitalisation duration in days (17.8 days versus 15.9 days; p = 0,051). Regarding mortality, 57 deaths were recorded in this study, representing 11.3% of live births. There was a significant difference in terms of neonatal mortality between the two groups, with a higher death rate in the group of children whose diagnosis was prenatal (17.2% versus 2.1%; p < 0.001). After excluding children who died for reasons other than CHD, there was no significant difference in terms of mortality in cases of pre- or postnatal diagnoses (56.6% versus 100%; p = 0,140). The same applies to the preoperative mortality of these children (56.7% versus 75%; p = 0,627).

## Discussion

The overall prenatal detection rate for CHD is 71.5% in the southern region of France. This rate is similar to that found in the study led by Durand et al. in Haute-Normandie [7]. In 2005, Khoshnood et al. [6] reported a prenatal detection rate of 47.3% in the Paris area, and the EUROCAT study in 2009 [1] reported detection rates ranging from 1% (Malta) to 42.5% (France). In Europe, such differences between countries can be explained by unequal access to

healthcare and by discrepancies in prenatal screening organisation policies. In France, this difference can be explained by the disparity in design between studies. In addition, improved detection rates could be explained by the evolving guidelines from the CNEOF (National Conference on Obstetrical and Fetal Ultrasound) and the French College of Foetal Sonography (CFEF), requiring three images of the foetal heart during ultrasounds performed in the second and third trimesters: the four cavities and the right and left ejection channels.

The prenatal diagnosis rate of the 3 groups of CHD varied considerably (97.8% in group 1, 63.6% in group 2 and 65.9% in group 3). In the study by Durand et al. [7], these rates were 93%, 53% and 77%, respectively. The high prenatal detection rate for the CHDs in group 1 can be explained by the significant disorganisation in cardiac architecture that they produce. Our study found rates comparable with those reported in the literature [6,16]. Among group 2, transposition of the great arteries (TGA) was detected in 80.8% of cases, which seems higher than the percentage in other studies (70%) [7,16,17]. However, one TGA out of 5 is not detected before birth, whereas this pathology requires specific neonatal management. It was proven by several studies that the diagnosis of TGA before birth improved the survival rates for these children [8,18–20]. A total of 56.3% of coarctation of the aorta cases were detected before birth, which is a higher rate than that of other studies [16,21]. This is a very difficult CHD to detect in the prenatal period as it forms after birth. Some studies have suggested that some cases of aortic coarctation would never be detected in prenatal screening [21]. Total anomalous pulmonary venous return (TAPVR) is a rare CHD that is difficult to detect with prenatal screening [22–24]. Our results reveal a low detection rate (20%), but it is higher than that of other studies [7,24]. Children with TAPVR are at risk of cardiac decompensation at birth in cases of a blocked TAPVR and require immediate surgical intervention. Among the CHDs associated with genetic risk, tetralogy of Fallot and common arterial truncus were prenatally identified in approximately 75% of cases and atrioventricular septal defects in more than 90% of cases. The detection rate for CHD seems high in Southern France but could be improved nevertheless. Some studies have demonstrated the value of training on the efficiency of prenatal heart disease screening [7,15].

The gestational age for a CHD diagnosis has improved since the 1980s, from approximately 27 to 23 weeks of gestation [6]. Before the 2000s, 35% of CHD cases were diagnosed in the third trimester [25]. In our study, 76.6% of CHD cases were detected in the second trimester, with an average gestational age of 23.7 weeks of gestation. In our study, 5.1% of CHDs were diagnosed in the first trimester. CNEOF and CFEF does not currently recommend performing image of fetal heart during ultrasound in the first trimester. To improve the rate of early diagnosis, ISUOG (International Society of Ultrasound in Obstetrics and Gynaecology) recommends verifying the symmetry of the 4 cavities during ultrasound screening in the first trimester [26]. Some studies include a systematic foetal heart analysis in the first trimester ultrasound, making it possible to diagnose or suspect 90% of the most serious CHD cases and 42% of the more minor ones [27].

Our results show that 16.1% of CHD cases were associated with a genetic anomaly, a rate comparable with that stated in the study by Cohen et al. [14]. De Groote [28] found a higher rate (25–40%) of genetic anomalies in cases of severe CHD. The main genetic anomalies identified were trisomy 21 (35.7%), trisomy 18 (19.5%) and microdeletion 22Q11 (15.6%). This association shows the importance of a precise prenatal diagnosis of CHD to orient the genetic screening process not to overlook a genetic anomaly whose diagnosis could lead to a TOP upon maternal request. Our study showed that a TOP was executed in 32.6% of CHD cases with a prenatal diagnosis and an average gestational age of realisation for TOP of 24.2 weeks for gestation. In France, the law allows TOP to be performed at any time during pregnancy when the unborn child suffers from a disease of particular gravity recognized as incurable at

the time of diagnosis [29]. The CNGOF (French National College of Gynecology-obstetric) recommends carrying out a feticide in situations where the gestational age is advanced with a high probability that the child will be born alive without spontaneous death envisaged in short term [30]. For these reasons, an early diagnosis of CHD would make it possible to perform these TOPs earlier and avoid feticide. In our cohort, the average gestational age of TOP at diagnosis in the first trimester was 16.2 weeks. Among groups 1 and 2, the severity of the CHD motivated the TOP decision then almost in group 3, it was the presence of an associated genetic anomaly that motivated this request in 53.7% of cases. In children born with CHD, the death rate was 11.3%. Conversely, in other studies [31], our study did not show any benefit from a prenatal diagnosis in terms of mortality in children who died from their CHD or in terms of preoperative mortality. Certain studies [32,33] have demonstrated a reduction in the preoperative morbidity of these children in terms of preoperative ventilation, administration of antibiotics, and emergency surgery. However, these factors were not investigated in our study.

The present study has several limitations. Although this study covered a large population, 677 patients, the rarity of certain CHDs makes it difficult to interpret the results. Furthermore, only children with severe CHD requiring hospitalisation were included. Children who died in outlying maternity clinics and who were not diagnosed before birth could also not be identified. However, they potentially represent a smaller number of cases. Unfortunately, the retrospective design of our study did not allowed us to analyze the causes of screening failures (the quality of the screening ultrasound images, level and grade of the initial sonographer, maternal BMI, a lack of follow-up). This study did not allow us to measure the impact of a prenatal diagnosis on neonatal morbidity. Some studies have shown that a prenatal diagnosis enables the improvement of the neurocognitive prognosis of children with CHDs [8,34,35].

This study made it possible to carry out an inventory of the prenatal screening of CHDs in our region. An undergoing training programme for professionals in the region began with the objective of improving the accuracy of prenatal screening for specific CHDs. A new screening assessment will be conducted in our region after the end of these training programs, including patient characteristics, screening ultrasound images.

In conclusion, the detection rate for CHD appears to be globally satisfactory in Southern France. However, it remains perfectible for certain CHDs, particularly those in group 2 requiring adapted neonatal management and for conotruncal malformations in group 3 with a risk of genetic involvement. A training programme for professionals in the region actually in progress might improve the accuracy of prenatal screening for specific CHDs and will be further assessed.

## Supporting information

**S1 File. Transposition of the great arteries: The keys to screening.** VD: Right ventricle; VG: Left ventricle; OD: Right atrium; OG: left atrium; VP: Pulmonary vein; Ao: Aorta; AP: Pulmonary artery; VCS: superior vena cava.
(PDF)

**S2 File. Total anomalous pulmonary venous returns: The keys to screening.** VD: Right ventricle; VG: Left ventricle; OD: Right atrium; OG: left atrium; VP: Pulmonary vein.
(PDF)

**S3 File. Aorta coarctation: The keys to screening.** VD: Right ventricle; VG: Left ventricle; VP: Pulmonary vein; Ao: Aorta; AP: Pulmonary artery; VCS: superior vena cava.
(PDF)

**S1 Data.**
(XLSX)

## Acknowledgments

We wish to thank U. Agarwal and J. SL Lim for their critical reading of the manuscript, We wish to thank Réseau Méditerranée PACA-Corse-Monaco and all patients.

## Author Contributions

**Conceptualization:** Cornélie Suard, Audrey Flori, Florence Bretelle.

**Data curation:** Cornélie Suard, Audrey Flori.

**Formal analysis:** Anderson Loundou.

**Investigation:** Cornélie Suard, Audrey Flori, Florence Bretelle.

**Methodology:** Cornélie Suard, Audrey Flori, Florent Paoli, Anderson Loundou, Claude D'Ercole, Florence Bretelle.

**Project administration:** Cornélie Suard, Virginie Fouilloux, Sabine Sigaudy, Fabrice Michel, Julie Antomarchi, Pamela Moceri, Véronique Paquis-Flucklinger, Claude D'Ercole.

**Resources:** Anderson Loundou.

**Supervision:** Florent Paoli, Virginie Fouilloux, Claude D'Ercole, Florence Bretelle.

**Validation:** Florent Paoli, Anderson Loundou, Claude D'Ercole, Florence Bretelle.

**Visualization:** Cornélie Suard, Audrey Flori, Claude D'Ercole, Florence Bretelle.

**Writing – original draft:** Cornélie Suard, Audrey Flori.

**Writing – review & editing:** Cornélie Suard, Florence Bretelle.

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
