## [Decision Letter · Decision Letter 0]

9 Jun 2020

PONE-D-20-14787

Accuracy of prenatal screening for congenital heart disease in population: A retrospective study in Southern France between January 2014 and December 2017

PLOS ONE

Dear Dr. Suard,

Thank you for submitting your manuscript to PLOS ONE. After careful consideration, we feel that it has merit but does not fully meet PLOS ONE’s publication criteria as it currently stands. Therefore, we invite you to submit a revised version of the manuscript that addresses the points raised during the review process.

Please pay close attention to the reviewers comments attached especially with regard to giving greater clarity to the reader to understand local guidelines and practice

We look forward to receiving your revised manuscript.

Kind regards,

Andrew Sharp, PhD

Academic Editor

PLOS ONE

Journal Requirements:

2.  In ethics statement in the manuscript and in the online submission form, please provide additional information about the patient records used in your retrospective study. Specifically, please ensure that you have discussed whether all data were fully anonymized before you accessed them and/or whether the IRB or ethics committee waived the requirement for informed consent. If patients provided informed written consent to have data from their medical records used in research, please include this information.

3. Thank you for including your ethics statement:

"This study received a favourable opinion from the Ethics Committee at the University of Aix-Marseille.

The CIL (Correspondant Informatique et Libertés) was made aware of the study complying with French law (reference: DSN_2018-07-27_7419)."

i) Please amend your current ethics statement to confirm that your named institutional review board or ethics committee specifically approved this study.

ii) Once you have amended this/these statement(s) in the Methods section of the manuscript, please add the same text to the “Ethics Statement” field of the submission form (via “Edit Submission”).

Additional Editor Comments (if provided):

Many thanks for your submission

please address the reviewers comments paying particular attention to providing greater clarity on how fetocide and screening is performed in France and what the CFEF guidelines dictate

Reviewers' comments:

Reviewer's Responses to Questions

**Comments to the Author**

1. Is the manuscript technically sound, and do the data support the conclusions?

Reviewer #1: Partly

Reviewer #2: Yes

2. Has the statistical analysis been performed appropriately and rigorously? 

Reviewer #1: Yes

Reviewer #2: Yes

3. Have the authors made all data underlying the findings in their manuscript fully available?

Reviewer #1: Yes

Reviewer #2: Yes

4. Is the manuscript presented in an intelligible fashion and written in standard English?

Reviewer #1: Yes

Reviewer #2: Yes

5. Review Comments to the Author

Reviewer #1: The authors present a retrospective study across 2 centres from south of France looking at prenatal screening accuracy rates from 2014-17. My comments are :

1. In title authors state that they wish to evaluate impact on 'pregnancy issues'. This however seems too broad and the article does not go into any specific pregnancy issues except the tabular outcomes. I would have expected 'maternal/obstetric' outcomes on reading 'pregnancy issues' as main objective. This does not seem to be remit of paper. Authors could perhaps clarify what issues they mean at outset

2. Fig 1 with legend seems to be placed inappropriately in text

3. Data collected from lines 148-152 reads duplicate

4. The indication for molecular screening (line 160-61) is not clearly interpretable the way it has been written

5. Reword amenorrhoea in line 319 to gestation or gestational age

6. Use 'gestation' in place of 'term' in text

7. The average gestation for TOP was 24.2 weeks. Authors state that earlier diagnosis allowed doing TOP without fetocide (line 319-20). This is in stark contrast to UK practice where fetocide cut off is 21+6 weeks. Can authors please explain the fetocide process and its cut off ?

8. Line 254- I think 'morbidity' should read as 'mortality'

9. Table 2- Intrauterin should read as Intrauterine

10. The discussion raises further questions, which occur intutively to mind of the reader

a) What was outcome for 5.1% cases that were detected during the first trimester , authors do not discuss this further in discussion. What were these cases are were these in a specific centre using a specific algorithm like ISUOG paper that authors quote? Were these in Group 1,2 or 3 ?

b) For TOPs and IUDs was there any postmortem data ?

c) For the prenatally detected lesions, what was the 'concordance rates' on postnatal diagnosis ? Were any diagnoses revised from Group 2 to 3 or vice versa after postnatal period

d) Authors do not report on Right aortic arch which is far common condition that double arch, was this not recorded ?

e) Do they mean AVSD ( partial and/or total ) when they say AV Canal ?

f) They say tight pulmonary stenosis is unlikely to need immediate neonatal cardiac input. Why such a case is not likely to need emergency PV dilatation or stent ? Should this not be Group 2?

g) What were the other genetic conditions detected on microarray apart from the 3 they mention

h) I would like to see the 'screening algorithm' used in two centres- was it same or different ? Did the authors compare the detection rates between two centres and if one was better than the other?

i) The skill, level and grade of person doing the screening needs some discussion- was this sonographer,general obstetrician with interest in ultrasound, fetal medicine sub-specialist or fetal cardiologist ? The missed cases in Group 1, were the prenatal images checked on the 3 scans these women would have had ?

j) Was maternal BMI looked at in relation to anomalies that were missed or were only diagnosed after birth ? This data would be there in viewpoint and should be extractable

k) What is the relevance of p value in Table 2 ? Which groups are being compared and discussion could be expanded to cover this section.

The overall level of discussion needs improving. At present it mostly reads as tables that have been padded out in sentences.

It does not reflect the 'critical thinking' behind why work was done, how it changed anything at the place of study and what needs to be done to make things better. A section on what is known and what the study adds also should be added.

There is hint of French English style of writing, the chief editor should decide if this is acceptable.

Reviewer #2: Overall good review of impressive database (covers whole of Southern France) with no previous similar publication for the same population.

1. I struggled to ascertain the screening protocol for this study. On lines 284 to 286, the author mentions that they use the French College of Foetal Sonography (CFEF) criteria which includes four cavities and the right and left ejection channels. Later on lines 308-310, the author mentions ISUOG guidelines. I was unable to find the actual CFEF cardiac screening guidelines, although there was a reference to the guidelines in a publication (https://www.cfef.org/fichiers/EF0501.pdf) with one reference cited in French.

It would be interesting to know when these guidelines were used from, and if there are any changes in the guidelines. It is remarkable that the antental detection rate for outflow tract anomalies are significantly higher than those published in the UK, despite less strict criteria used (ie no 3VT or 3VV views used). One wonders if the French sonographers are better trained than UK sonographers?

2. One of the references (nos 2 with reference to a EUROCAT review on CHD) has been incorrectly referenced. I could not access the page according to the webpage address given. It should have been referenced as a Circulation 2011;123:841-849 article.

3. In table 3 (outcomes of children with CHD), there were data on death before surgery. It is unclear if it is death due to cardiac issues before the surgery or death as a result of compassionate care.

Overall good review of data.

6. PLOS authors have the option to publish the peer review history of their article (what does this mean?). If published, this will include your full peer review and any attached files.

Reviewer #1: Yes: Umber Agarwal

Reviewer #2: Yes: Joyce SL Lim

---

## [Author Response · Author response to Decision Letter 0]

26 Jul 2020

Response to Reviewers

Dear Doctor Agarwal and Doctor Lim,

Please find below our answers to each reviewers’ questions and comments regarding the manuscript of our latest article entitled ‘Accuracy of prenatal screening for congenital heart disease in population: A retrospective study in Southern France’. We are grateful to have the opportunity to submit this new and revised version of the manuscript. We carefully read Reviewers’ comments and attempted to answer each of them. 

Please find enclosed the revised and the point to point answer.

Yours sincerely, 

The Authors

Reviewer #1: 

The authors present a retrospective study across 2 centres from south of France looking at prenatal screening accuracy rates from 2014-17. My comments are :

1. In title authors state that they wish to evaluate impact on 'pregnancy issues'. This however seems too broad and the article does not go into any specific pregnancy issues except the tabular outcomes. I would have expected 'maternal/obstetric' outcomes on reading 'pregnancy issues' as main objective. This does not seem to be remit of paper. Authors could perhaps clarify what issues they mean at outset

We agree with this comment. The objective of this study is to assess the impact of the diagnosis of congenital heart disease (CHD) on whether or not pregnancy continues depending on the severity of CHD. The objective is not to assess the obstetric impact of these fetal malformations diagnosis.

Thus, the term “issue” has been replaced by the term “outcome” to clarify.

Lines 40, 79

2. Fig 1 with legend seems to be placed inappropriately in text

Thank you for this remark. The reference to the figure 1 has been moved to line 102.

3. Data collected from lines 148-152 reads duplicate

May I ask that you re-confirm that the duplicate data ‘lines 148-152’ was:

‘In cases of a postnatal diagnosis, the parameters studied were the age at the time of diagnosis and the discovery of an associated PMS or genetic anomaly. For all children born with CHD, the following data were collected: the requirement for hospitalisation in the neonatal intensive care unit and the duration in number of days, the existence of PMS or an associated genetic anomaly, and the occurrence of death.’

4. The indication for molecular screening (line 160-61) is not clearly interpretable the way it has been written

Following your advice, the changes were made in the manuscript: Amniocentesis with karyotype or CGH array was proposed for all patient in case of prenatal diagnosis. 

Lines 130-131.

5. Reword amenorrhoea in line 319 to gestation or gestational age

Thank you for this remark. Correction is made on line 293

6. Use 'gestation' in place of 'term' in text

All over the manuscript ‘gestational age’ used in place ‘term’. (lines 120, 177, 293, 306)

7. The average gestation for TOP was 24.2 weeks. Authors state that earlier diagnosis allowed doing TOP without fetocide (line 319-20). This is in stark contrast to UK practice where fetocide cut off is 21+6 weeks. Can authors please explain the fetocide process and its cut off ?

Thank you, this is very relevant. Therefore, tried to clarify and detail these different points in the text.

A) Line 319-320: “An early diagnosis of CHD made it possible to perform these TOPs earlier and avoid feticide”.

What we meant is that an early diagnosis of CHD would make it possible to perform a TOP earlier, which would allow us to avoid the realization of a feticide which can be difficult for the couple. The correction to the text was performed in lines 319-320.

In the text: “An early diagnosis of CHD would make it possible to perform these TOPs earlier and avoid feticide”.

B) Regarding the fetocide process and its cut off, 

- According to French law of the public health code, article L2213-1: ‘Voluntary termination of pregnancy can, at any time, be practiced if two doctors members of a multidisciplinary medical team attest, after opinion, either that the continuation of pregnancy puts in serious danger the health woman, or that there is a strong probability that the unborn child is affected by a affection of a particular gravity recognized as incurable in time of diagnosis’.

- The CNGOF (French National College of Gynecology-Obstetric) therefore issued recommendations on 3.12.2008 on the management of these terminations depending on the gestational age. He recommends: ‘the feticide will be produced in situations where the gestational age is advanced with a high probability that the child will be born alive without spontaneous death envisaged in the short term. The gestational age of 24 weeks is used by most teams’.

- In France, most teams realizes a feticide for TOP from 22 weeks. We wanted to express that the early diagnosis of CHD allowed if the request was made by the couple to carry out an early TOP before 22 weeks which made it possible to avoid feticide.

In the manuscript, we modified the sentences to: ‘In France, the law allows TOP to be performed at any time of pregnancy when the unborn child suffers from a disease of particular severity recognized as incurable at the time of diagnosis �29�. The CNGOF (French National College of Gynecology-obstetric) recommends carrying out a feticide in situations where the gestational age is advanced with a high probability that the child will be born alive without spontaneous death envisaged in short term �30�. For these reasons, an early diagnosis of CHD would make it possible to perform these TOPs earlier and avoid feticide’. Lines 305-312

8. Line 254- I think 'morbidity' should read as 'mortality'

Thank you for this remark. The term ‘morbidity’ has been replaced by the term ‘mortality’ in the line 242.

9. Table 2- Intrauterin should read as Intrauterine

The correction was made in the table 2.

10. The discussion raises further questions, which occur intutively to mind of the reader

a) What was outcome for 5.1% cases that were detected during the first trimester , authors do not discuss this further in discussion. What were these cases are were these in a specific centre using a specific algorithm like ISUOG paper that authors quote? Were these in Group 1,2 or 3 ?

The CNEOF (National Conference on Obstetrical and Fetal Ultrasound) and the CFEF (French College of Foetal Sonography) do not currently recommend performing image of fetal heart examination during first trimester screening ultrasound. All CHDs in this study were assessed in one of three Multidisciplinary Prenatal Diagnosis Centres applying recommendations of the CNEOF and CFEF for diagnostic ultrasound (http://www.cfef.org/archives/bricabrac/cnte-echodiag.pdf).

We understand early CHD diagnosis is a the goal, therefore we added in the manuscript the description of this subgroup:

In our study, only 5.1% of CHDs were diagnosed in the first trimester. Among the 25 CHDs diagnosed in the first trimester, 13 were from group 1, 1 from group 2, 11 from group 3. Regarding the outcome of these pregnancies: 18 TOPs were performed at an average gestational age 16.2 weeks, 3 IUDs were observed, and 4 children were born and had surgery.

Lines 190-183 in results section.

Lines 294 and 312 in discussion section.

b) For TOPs and IUDs was there any postmortem data ?

We have recorded whether an autopsy had been performed. The correlation between prenatal diagnosis and autopsy was not the objective of the present study. An autopsy was performed in 35.4% of cases of TOPs and in 11.7% of cases of IUDs. 

Thank you for your comment, this was add in the result section. Lines 208-209

c) For the prenatally detected lesions, what was the 'concordance rates' on postnatal diagnosis ? Were any diagnoses revised from Group 2 to 3 or vice versa after postnatal period

Among the children borned alive with a CHD diagnosed prenatally, only thirteen diagnoses were reversed. These thirteen cases were prenatal suspicions of coarctation of the aorta that did not occur at birth. All other diagnosis were postnatally confirmed.

Thus, no CHDs from Group 2 to 3 or vice versa were revised after postnatal period.

Thank you for your comment, this was add in the result section. Lines 209-211

d) Authors do not report on Right aortic arch which is far common condition that double arch, was this not recorded ?

It’s true that, right aortic arch is more common than double one. For the study right aortic arch was considered as an anatomic variation if isolated. The right aortic arch were not recorded into this study.

e) Do they mean AVSD ( partial and/or total ) when they say AV Canal ?

Thank you for this remark.

Yes, we mean ‘atrioventricual septal defect’ when we used ‘atrioventricular canal’

We have correct this term in the text (line 181, Table 1, line 302)

f) They say tight pulmonary stenosis is unlikely to need immediate neonatal cardiac input. Why such a case is not likely to need emergency PV dilatation or stent ? Should this not be Group 2?

Tight pulmonary stenosis has been classified as group 3. When they were classified as critical pulmonary stenosis with risk of urgent neonatal management, they were classified in group 2 as shown in figure and table 1. 

g) What were the other genetic conditions detected on microarray apart from the 3 they mention?

The most frequently found genetic anomalies were trisomy 18, 21 and microdeletion 22q11 (197-199). The other anomalies found were trisomy 13, triploidy, deletion chromosome 4,5,6,7,8,10, 17,18, Turner syndrome, duplication chromosome X, duplication chromosome 8, partial trisomy chromosome 11 and 22.

h) I would like to see the 'screening algorithm' used in two centres- was it same or different ? Did the authors compare the detection rates between two centres and if one was better than the other?

In 2 centers, screening algorithm were the same (CNEOF and CFEF recommendations). The data were analyzed from the medical records in the same way. 

We recorded all the fetuses expertized in the MPCDs and all the children under the age of one having been hospitalized in the cardiology departments of the UTHs of Nice and Marseille. 

We did not intend to compare the two centers because the objective of this study was to evaluate the overall regional screening of CHDs. In addition, there is a common perinatal network in our region (Mediterranean network: PACA-Corsica-Monaco) which organized common training courses for the sonographers of the region allowing a homogenization of practices.

The center detection rate was not available because Marseille is the surgical CHD reference center and in case of postnatal diagnosis children can be referred from the whole region. Thus was not calculated this rate by center. 

To clarified our results, you will find above the modified study flow chart (Fig 2).

Figure 2. Flow chart

i) The skill, level and grade of person doing the screening needs some discussion- was this sonographer, general obstetrician with interest in ultrasound, fetal medicine sub-specialist or fetal cardiologist ? The missed cases in Group 1, were the prenatal images checked on the 3 scans these women would have had ?

A) Due to the retrospective nature of this study, we were unable to collect the level and grade of the initial sonographer (Midwife, radiologist, medical gynecologist, obstetrician gynecologist). Once referred one of the three MPDCs all the CHDs were always assessed by an ultrasound specialist in fetal medicine and by a cardiopediatrician.

B) No, unfortunately in case of post natal diagnosis we didn’t have access to ultrasound images. As our study was retrospective at the beginning of it, we try to collect the screenind images but it was not possible. Thus this very interesting data is missing, we add this limitation in the discussion section. Lines 324-326….

‘Unfortunately, the retrospective design of our study did not allowed us to analyze the cause of screening failures (the quality of the screening ultrasound images, level and grade of the initial sonographer, maternal BMI, a lack of follow-up).’

j) Was maternal BMI looked at in relation to anomalies that were missed or were only diagnosed after birth ? This data would be there in viewpoint and should be extractable

In the same way as you, we wanted to study the impact of a high BMI on screening failure. Unfortunately this data was too rarely entered in our software to be usable on all data.

k) What is the relevance of p value in Table 2 ? Which groups are being compared and discussion could be expanded to cover this section.

Thank you for your useful comment. Table 2 is a descriptive table of pregnancy outcomes in case of prenatal diagnosis and not a comparison table. The p value has therefore been deleted for less confusion.

The overall level of discussion needs improving. At present it mostly reads as tables that have been padded out in sentences.

It does not reflect the 'critical thinking' behind why work was done, how it changed anything at the place of study and what needs to be done to make things better. A section on what is known and what the study adds also should be added.

Based on your advice, we have improved the discussion. We have completed the limitations of the study. We have also completed the impact of this study on the implementation of training programs in our region to improve the screening of CHDs. Lines 329-332

There is hint of French English style of writing, the chief editor should decide if this is acceptable.

We have had the whole article translated and corrected twice. Invoices for these translations are sent to the publisher.

Reviewer #2: 

Overall good review of impressive database (covers whole of Southern France) with no previous similar publication for the same population.

1. I struggled to ascertain the screening protocol for this study. 

On lines 284 to 286, the author mentions that they use the French College of Foetal Sonography (CFEF) criteria which includes four cavities and the right and left ejection channels. Later on lines 308-310, the author mentions ISUOG guidelines. I was unable to find the actual CFEF cardiac screening guidelines, although there was a reference to the guidelines in a publication (https://www.cfef.org/fichiers/EF0501.pdf) with one reference cited in French.

It would be interesting to know when these guidelines were used from, and if there are any changes in the guidelines. 

It is remarkable that the antenatal detection rate for outflow tract anomalies are significantly higher than those published in the UK, despite less strict criteria used (ie no 3VT or 3VV views used). One wonders if the French sonographers are better trained than UK sonographers?

Thank you very much for these pertinent comments.

- Screening protocol: 

Following your remarks we have modified the flow chart of the study for more clarity (Fig 2) . 

We counted for this purpose all the fetuses expertized in the MPCDs and all the children under the age of one having been hospitalized in the cardiology departments of the UTHs of Nice and Marseille. 

Figure 2. Flow chart

- Indeed, the CNEOF and the CFEF currently recommends in France to carry out images of four cavities as well as the right and left ejection channels during the second and third trimester ultrasound. These recommendations are not found with regard to the first trimester screening ultrasound. This is why we later quote the ISUOG guidelines which proposed these recommendations in the first trimester. 

lines 294-295.

- We will send you the link for the CFEF 2016 guideline on screening ultrasound: www.cfef.org/archives/bricabrac/cneof/compte-renducneof2016.pdf 

- We don't know why our screening rates appear to be higher than those published in the UK. Perhaps there is a difference in terms of the number of screening ultrasounds recommended? The 3VV cut is indeed most often performed in daily screening practice. However, it is included in our recommendations for diagnostic ultrasound, not screening ultrasound.

2. One of the references (nos 2 with reference to a EUROCAT review on CHD) has been incorrectly referenced. I could not access the page according to the webpage address given. It should have been referenced as a Circulation 2011;123:841-849 article.

Thank you for this comment. This error has been corrected

3. In table 3 (outcomes of children with CHD), there were data on death before surgery. It is unclear if it is death due to cardiac issues before the surgery or death as a result of compassionate care.

It’s true that these two situations have not been differentiated. Deaths secondary to CHDs in groups 2 and 3 occurred before curative surgery was performed with no initial compassionate care. For the CHDs in group 1, this information is not available in case of death before than palliative surgery was envisaged.

Overall good review of data.

We are very grateful for the opportunity to improve our manuscript. 

The Authors.

---

## [Decision Letter · Decision Letter 1]

7 Sep 2020

Accuracy of prenatal screening for congenital heart disease in population: A retrospective study in Southern France

PONE-D-20-14787R1

Dear Dr. Suard,

We’re pleased to inform you that your manuscript has been judged scientifically suitable for publication and will be formally accepted for publication once it meets all outstanding technical requirements.

Kind regards,

Andrew Sharp, PhD

Academic Editor

PLOS ONE

Additional Editor Comments (optional):

please correct the spelling of Truncus as mentioned in the second review

Reviewers' comments:

Reviewer's Responses to Questions

**Comments to the Author**

1. If the authors have adequately addressed your comments raised in a previous round of review and you feel that this manuscript is now acceptable for publication, you may indicate that here to bypass the “Comments to the Author” section, enter your conflict of interest statement in the “Confidential to Editor” section, and submit your "Accept" recommendation.

Reviewer #1: All comments have been addressed

2. Is the manuscript technically sound, and do the data support the conclusions?

Reviewer #1: Yes

3. Has the statistical analysis been performed appropriately and rigorously? 

Reviewer #1: Yes

4. Have the authors made all data underlying the findings in their manuscript fully available?

Reviewer #1: Yes

5. Is the manuscript presented in an intelligible fashion and written in standard English?

Reviewer #1: Yes

6. Review Comments to the Author

Reviewer #1: I am grateful to authors for a detailed revision of their manuscript and addressing all our concerns. The manuscript is now looking very good and is of high quality and with clear messages. I am very pleased

One final spelling mistake to be corrected - troncus should be replaced with truncus !

7. PLOS authors have the option to publish the peer review history of their article (what does this mean?). If published, this will include your full peer review and any attached files.

Reviewer #1: **Yes: **Mr Umber Agarwal

---

## [Editor Report · Acceptance letter]

14 Sep 2020

PONE-D-20-14787R1 

Accuracy of prenatal screening for congenital heart disease in population: A retrospective study in Southern France 

Dear Dr. SUARD:

I'm pleased to inform you that your manuscript has been deemed suitable for publication in PLOS ONE. Congratulations! Your manuscript is now with our production department. 

Kind regards, 

on behalf of

Dr. Andrew Sharp 

Academic Editor

PLOS ONE